# A large effective population size for established within-host influenza virus infection

Casper K Lumby[1], Lei Zhao[1], Judith Breuer[2,3], Christopher JR Illingworth[1,4,5]*

[1]Department of Genetics, University of Cambridge, Cambridge, United Kingdom; [2]Great Ormond Street Hospital, London, United Kingdom; [3]Division of Infection and Immunity, University College London, London, United Kingdom; [4]Department of Applied Mathematics and Theoretical Physics, University of Cambridge, Cambridge, United Kingdom; [5]Department of Computer Science, Institute of Biotechnology, University of Helsinki, Helsinki, Finland

**Abstract** Strains of the influenza virus form coherent global populations, yet exist at the level of single infections in individual hosts. The relationship between these scales is a critical topic for understanding viral evolution. Here we investigate the within-host relationship between selection and the stochastic effects of genetic drift, estimating an effective population size of infection $N_e$ for influenza infection. Examining whole-genome sequence data describing a chronic case of influenza B in a severely immunocompromised child we infer an $N_e$ of $2.5 \times 10^7$ (95% confidence range $1.0 \times 10^7$ to $9.0 \times 10^7$) suggesting that genetic drift is of minimal importance during an established influenza infection. Our result, supported by data from influenza A infection, suggests that positive selection during within-host infection is primarily limited by the typically short period of infection. Atypically long infections may have a disproportionate influence upon global patterns of viral evolution.

*For correspondence:
cjri2@cam.ac.uk

Competing interests: The authors declare that no competing interests exist.

## Introduction

The evolution of the influenza virus may be considered across a broad range of scales. On a global level, populations exhibit coherent behaviour (*Buonagurio et al., 1986*; *Fitch et al., 1997*; *Bedford et al., 2015*), evolving rapidly under collective host immune pressure (*Ferguson et al., 2003*; *Grenfell et al., 2004*). On another level, these global populations are nothing more than very large numbers of individual host infections, separated by transmission events.

Despite the clear role for selection in global influenza populations, recent studies of within-host infection have suggested that positive selection does not strongly influence evolution at this smaller scale (*Debbink et al., 2017*; *McCrone et al., 2018*; *Han et al., 2019*). Contrasting explanations have been given for this, with suggestions either that selection at the within-host level is intrinsically inefficient, being dominated by stochastic processes (*McCrone et al., 2018*), or that while selection is efficient, a mismatch in timing between the peak viral titre and the host adaptive immune response prevents selection from taking effect (*Han et al., 2019*).

To resolve this issue, we evaluated the relative importance of selection and genetic drift during a case of influenza infection. The balance between these factors is determined by the effective size of the population, denoted $N_e$. If $N_e$ is high, selection will outweigh genetic drift, even where differences in viral fitness are small (*Rouzine et al., 2001*). By contrast, if $N_e$ is low, less fit viruses are more likely to outcompete their fitter compatriots.

Estimating $N_e$ is a difficult task, with a long history of method development in this area (*Wright, 1938*; *Wang et al., 2016*; *Khatri and Burt, 2019*). A simple measure of $N_e$ may be

calculated by matching the genetic change in allele frequencies in a population with the changes occurring in an idealised population evolving under genetic drift (*Kimura and Crow, 1963*). However, such estimates are vulnerable to distortion, for example being reduced by the effect of positive selection in a population. Where the global influenza A/H3N2 population is driven by repeated selective sweeps (*Fitch et al., 1991*; *Rambaut et al., 2008*; *Strelkowa and Lässig, 2012*) a neutral estimation method suggests a value for $N_e$ not much greater than 100 (*Bedford et al., 2010*). While methods for jointly estimating $N_e$ and selection exist, they are limited in considering only a few loci in linkage disequililbrium (*Bollback et al., 2008*; *Feder et al., 2014*; *Foll et al., 2014*; *Terhorst et al., 2015*; *Rousseau et al., 2017*). Non-trivial population structure can affect estimates (*Laporte and Charlesworth, 2002*); a growing body of evidence supports the existence of structure in within-host influenza infection (*Lakdawala et al., 2015*; *Sobel Leonard et al., 2017a*; *Richard et al., 2018*; *Hamada et al., 2012*). While careful experimental techniques can reduce sequencing error (*McCrone and Lauring, 2016*), noise from sequencing and unrepresentative sample collection combine (*Illingworth et al., 2017*), potentially confounding estimates of $N_e$ in viral populations (*Lumby et al., 2018*). If $N_e$ is high, any signal of drift can be obscured by noise.

We here estimate a mean effective population size for an established within-host influenza B infection using data collected from a severely immunocompromised host. While the viral load of the infection was not unusual for a hospitalised childhood infection (*Wishaupt et al., 2017*), an absence of cell-mediated immunity led to the persistence of the infection for several months (*Lumby et al., 2020*). Given extensive sequence data collected during infection, the reduced role of positive selection, combined with novel methods to account for noise and population structure, enabled an improved inference of $N_e$. The large effective size we infer suggests that selection acts in an efficient manner during an established influenza infection. Even in more typical cases, the influence of positive selection is likely to be limited only by the duration of infection.

## Results and discussion

Viral samples were collected at 41 time points spanning 8 months during the course of an influenza B infection in a severely immunocompromised host (*Figure 1A*). Clinical details of the case, and the use of viral sequence data in evaluating the effectiveness of clinical intervention, have been described elsewhere (*Lumby et al., 2020*). After unsuccessful treatment with oseltamivir, zanamivir and nitazoxanide, a bone marrow transplant and favipiravir combination therapy led to the apparent clearance of infection. Apart from a single exception, biweekly samples tested negative for influenza across a period of close to two months. A subsequent resurgence of zanamivir-resistant infection was cleared by favipiravir and zanamivir in combination.

Phylogenetic analysis of whole-genome viral consensus sequences showed the existence of non-trivial population structure, with at least two distinct clades emerging over time (*Figure 1B*, *Figure 1—figure supplement 1*); we term these clades A and B. Having diverged, the two clades persisted across several months of infection. Haplotype reconstruction showed that samples from clade B were comprised of distinct viral haplotypes to those from clade A; similar patterns were observed in different viral segments (*Figure 1—figure supplement 2*). The October 4th sample is intermediate between the initial and final samples collected (*Figure 1D*). We suggest that, from a common evolutionary origin, Clade B slowly evolved away from the initial consensus, while viruses in clade A stayed close in sequence space to this consensus. The cladal structure suggests the existence of spatially distinct viral populations in the host, samples stochastically representing one population or the other.

To estimate the effective population size, we analysed genome-wide sequence data from samples in clade A collected before first use of favipiravir. A method of linear regression was used to quantify the rate of viral evolution, measuring the genetic distance between samples as a function of increasing time between dates of sample collection. We inferred a rate equivalent to 0.051 substitutions per day (97.5% confidence interval 0.034 to 0.068) (*Figure 2A*), equivalent to 7.94 substitutions genome-wide across 157 days of evolution. The vertical intercept of this line provides an estimate of the contribution of noise to the measured distance between samples, potentially arising from sequencing error or undiagnosed population structure. The identified value of close to 40 substitutions is equivalent to a between-sample allele frequency difference of approximately +/- 0.3% per

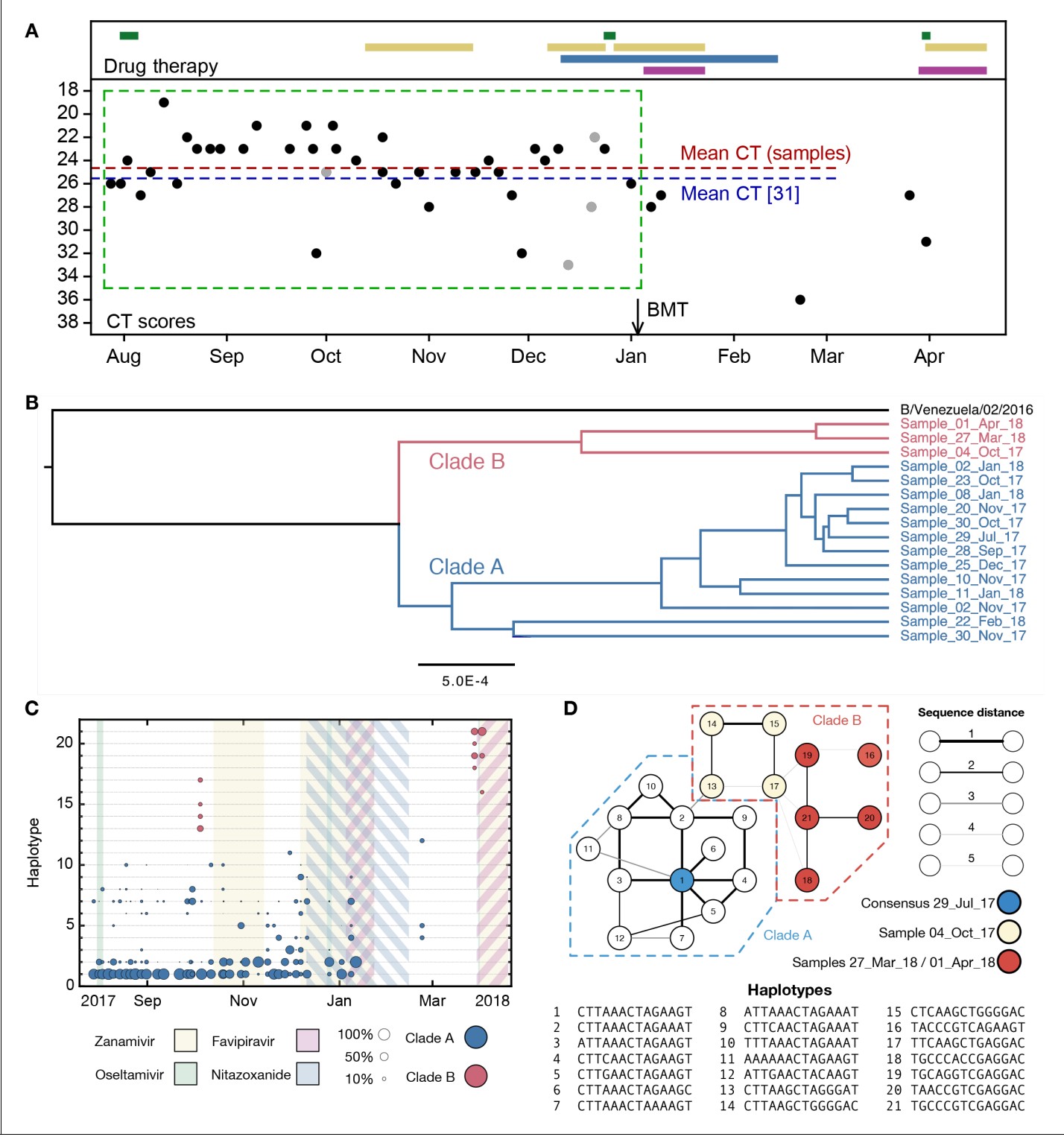

**Figure 1.** Population structure of the influenza infection. (**A**) CT values from viral samples collected over time indicate the viral load of the infection; a higher number corresponds to a lower viral load. Drug information, above, shows the times during which oseltamivir (green), zanamivir (yellow), nitazoxanide (blue) and favipiravir (purple) were prescribed. Black dots show samples from which viral sequence data were collected; gray dots show samples from which viral sequence data were not collected. The green box shows the window of time over which samples were analysed, preceding the use of favipiravir in January. The mean viral load (dashed horizontal line, red) was close to the mean reported for a set of samples from hospitalised children with influenza (dashed horizontal blue line) (**Wishaupt et al., 2017**). A black arrow shows the date of a bone marrow transplant (BMT). (**B**) A phylogeny of whole-genome viral consensus sequences identified two distinct clades in the viral population. Clade B featured three samples,

*Figure 1 continued on next page*

*Figure 1 continued*

distributed across the period of infection, with the remaining samples contained in Clade A. (C) Sub-consensus structure of the viral population inferred via a haplotype reconstruction algorithm using data from the neuraminidase segment. The same division of sequences into two clades is visible, with samples being comprised of distinct viral genotypes. The area of each circle is proportional to the inferred frequency of the corresponding haplotype in the viral population. Haplotypes reaching a frequency of at least 10% in at least one time point are shown. Multiple drugs were administered to the patient through time, with a favipiravir/zanamivir combination first causing a temporary reduction of the population to undetectable levels, then finally clearing the infection. Haplotypes spanned the loci 96, 170, 177, 402, 403, 483, 571, 653, 968, 973, 1011, 1079, 1170, and 1240 in the NA segment. (D) Evolutionary relationship between the haplotypes; clade B is distinct from and evolves away from those sequences comprising the initial infection. Numbers refer to the distinct haplotypes identified within the population.

The online version of this article includes the following source data and figure supplement(s) for figure 1:

**Source data 1.** Viral load and details of treatment with inferred haplotype frequencies for the neuraminidase viral segment.
**Source data 2.** Data for the phylogenetic tree in *Figure 1B*.
**Figure supplement 1.** Complete phylogeny of whole-genome viral consensus sequences, coloured by clade.
**Figure supplement 2.** Haplotype reconstruction for data describing the haemagglutinin segment of the virus.
**Figure supplement 2—source data 1.** Reconstructed haplotype frequencies for the haemagglutinin viral segment.

locus. While considerable noise affects each sample, the dataset as a whole provides a clear signal of evolutionary change.

A simulation based analysis, measuring the extent of evolution in idealised Wright-Fisher populations (*Kimura and Crow, 1963*), inferred an effective population size of $2.5 \times 10^7$ (95% confidence range $1.0 \times 10^7$ to $9.0 \times 10^7$) for viruses in clade A before the use of favipiravir (*Figure 2B*). This value is substantially larger than estimates made recently for within-host HIV infection (*Pennings et al., 2014*; *Rouzine et al., 2014*), and suggests that even weak selection could easily overcome genetic drift. Data from clade B gave a lower estimated value of $2 \times 10^6$, (95% confidence range $4 \times 10^5$ to $2 \times 10^8$) perhaps reflecting the less frequent observation of samples in that clade (*Figure 2C,D*), and the bottleneck induced by favipiravir, which was spanned by the data used in this calculation.

Our value of $N_e$ is representative of the population after the initial establishment of infection; the initial expansion of the viral population was not represented in our data. Population structure during the infection might have lowered the value we obtain (*Whitlock and Barton, 1997*). The partial onset of zanamivir resistant alleles (*Jackson et al., 2005*), sporadically observed at intermediate frequency in clade A after the administration of the drug (*Figure 2—figure supplement 1*), is suggestive of sampling a random mixture of viruses from resistant and susceptible subpopulations.

Our method equates change in a population with genetic drift (*Kimura and Crow, 1963*), neglecting the role of selection. As such, the influence of positive selection might have led us to underestimate $N_e$. While viral evolution was generally not driven by selection (*Figure 2—figure supplement 2*), positive selection (e.g. for zanamivir resistance) would increase the rate of viral evolution, lowering our inferred value. Selection may have influenced the division between clades, perhaps through the adaptation of the virus to specific local environments. Purifying selection may also have influenced the population in ways not accounted for by our method. Yet our result is clear. Once an infection is established, selection will dominate the stochastic effects of drift upon within-host evolution.

The dataset we considered is particularly suited to our calculation. The long period of infection combined with frequent sampling allowed for the characterisation of a slow rate of evolution amidst population structure and noise in the data. Further, the absence of strong selection reduced the error in our inference approach, which assumed an idealised neutral population. To provide further validation we repeated our approach on data describing long-term influenza A/H3N2 infection in four immunocompromised adults (*Xue et al., 2017*). The estimates for $N_e$ we obtained, of between $3 \times 10^5$ and $1 \times 10^6$ (*Figure 2—figure supplement 3*), while high, were smaller than for our flu B case, potentially being reduced by an increased influence of selection.

We believe that our study provides a first realistic estimate of within-host effective population size for severe influenza infection in humans. The viral load in the influenza B case was high, representative of hospitalised cases of childhood influenza infection. However, the magnitude of our inferred effective size, of order $10^7$, suggests that selection will predominate over drift even in more typical cases. Mean CT values for influenza in non-hospitalised children have been reported as

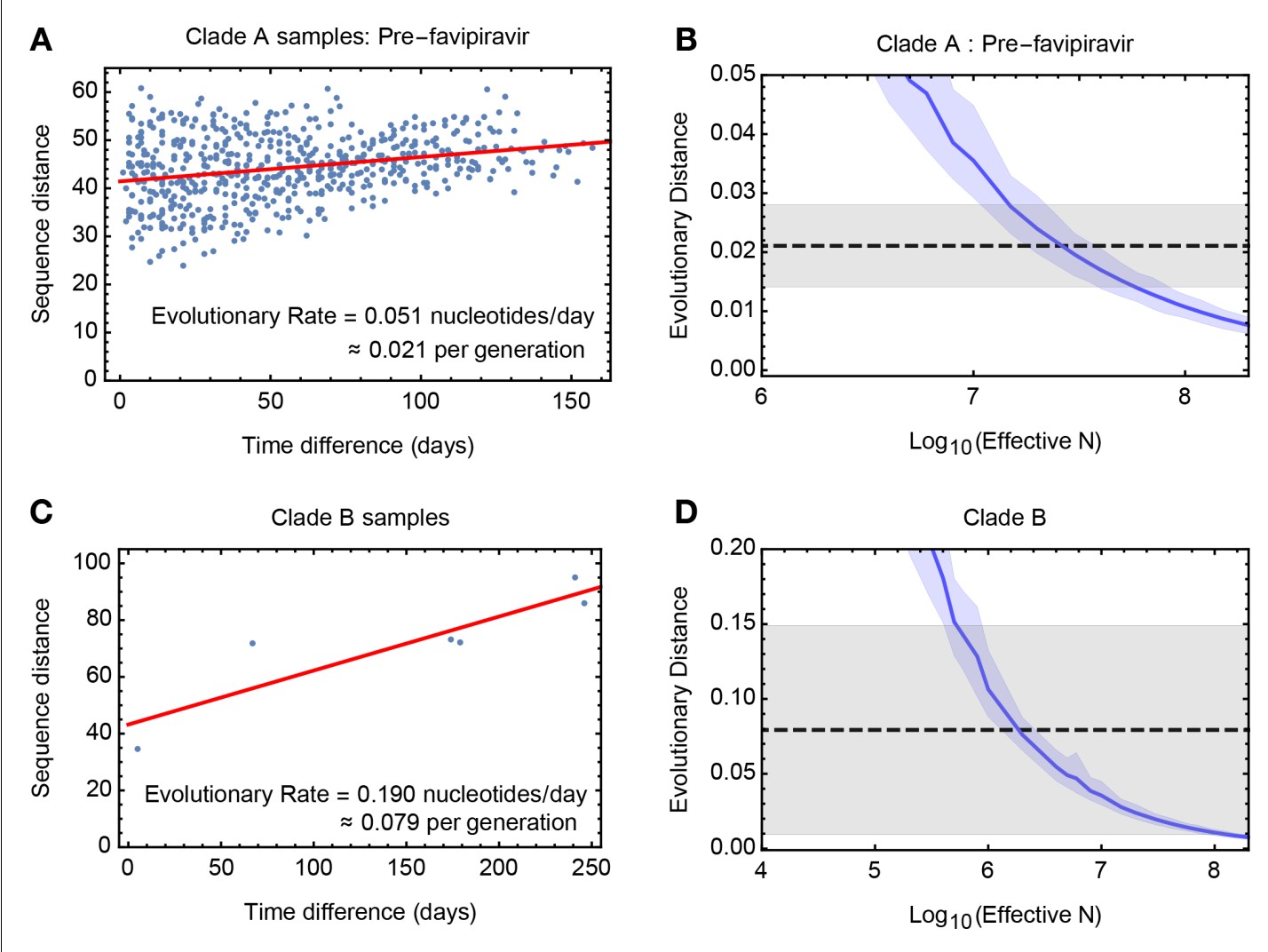

**Figure 2.** Measuring rates of evolution in the viral population. (**A**) Computed rate of evolution for viruses in clade A up to the time of the first use of favipiravir. The distance between two sequences is calculated as the total absolute difference in four-allele frequencies measured across the genome. The calculated rate per generation is based upon a generation time for influenza of 10 hours (*Nobusawa and Sato, 2006*). (**B**) Distribution of evolutionary distances in influenza populations simulated using a Wright-Fisher model compared to the distance per generation calculated in the regression fit. A solid blue line shows the mean, with shading indicating an approximate 97.5% confidence interval around the mean. Statistics were calculated from sets of 400 simulations conducted at each value of $N_e$. The dashed black line shows the rate of evolution of the real population; gray shading shows a 97.5% confidence interval for this statistic. (**C**) Calculated rate of evolution for viruses in clade B. For the purposes of calculating a rate of evolution the first sample collected from the patient was included as part of clade B. (**D**) Estimation of $N_e$ for clade B. The results of simulations shown here are identical to those in part B of the figure.

The online version of this article includes the following source data and figure supplement(s) for figure 2:

**Source data 1.** Between sample differences and simulated rates of evolution for clades A and B of the viral population.
**Figure supplement 1.** Amino acids present at codon 117 of the neuraminidase segment of the virus after the first administration of zanamivir.
**Figure supplement 1—source data 1.** Amino acid frequencies at position 117 in the neuraminidase viral segment.
**Figure supplement 2.** Rates of evolutionary change at non-synonymous and synonymous sites.
**Figure supplement 2—source data 1.** Synonymous and non-synonymous sequence distances calculated per nucleotide across the whole viral genome for different pairs of samples.
**Figure supplement 3.** Estimates of the effective population size for data from a study of long-term influenza A/H3N2 infection in four patients.
**Figure supplement 3—source data 1.** Sequence distances D calculated for the Xue et al dataset.
**Figure supplement 4.** Minority allele frequencies from distinct time points used for the Wright-Fisher simulation applied to the influenza B sequence data.
**Figure supplement 4—source data 1.** Sorted allele frequencies collected genome-wide for samples used in the simulation of data.

*Figure 2 continued on next page*

**Figure supplement 5.** Frequencies of minority variant alleles identified in the HCV01 dataset used to evaluate the accuracy of variant calling in our sequencing pipeline.

**Figure supplement 5—source data 1.** Replicate allele frequencies from the HCV01 dataset, described in a previous publication, and used in this study to estimate a frequency-dependent positive predictive value for variant calling using the sequencing method applied to the influenza B data.

**Figure supplement 6.** Regions of frequency space used to define observations and non-observations of allele frequencies.

**Figure supplement 7.** Positive predictive value for minority variants under our sequencing pipeline, calculated at different frequency ranges.

**Figure supplement 7—source data 1.** Frequency-dependent positive predictive values for variant calling.

around 10 units lower than those for hospitalised cases (*Wishaupt et al., 2017*); an order of magnitude calculation suggests an Ne, upon the establishment of infection, of approximately $10^4$ in such cases. Such a value again reflects an established population, not accounting for the initial population bottleneck. It has the implication that the evolution of a measurable variant (i.e. at a frequency of 1% or above) will be dominated by selection of a magnitude of 1% or greater per generation (*Rouzine et al., 2001*).

Our result supports the idea that a tight transmission bottleneck (*McCrone et al., 2018*; *Valesano, 2020*; *Ghafari et al., 2020*) followed by a short period of infection is sufficient to explain the observed lack of within-host variation in typical cases of influenza (*Debbink et al., 2017*; *McCrone et al., 2018*); the stochastic effects of genetic drift do not limit the impact of positive selection. Variants arising through de novo mutation would require strong selection to reach a substantial frequency during infection (*Zhao et al., 2019*), particularly if the onset of selection is delayed (*Miao et al., 2010*; *Illingworth et al., 2014*; *Morris, 2020*). We suggest that, while not being confounded by drift, selection does not usually have time to fix novel variants in the population, exceptions including the emergence of antiviral resistance and some cases of longer infection (*Xue et al., 2017*; *Gubareva et al., 1998*; *Snydman, 2006*; *Centers for Disease Control and Prevention (CDC), 2009*; *Imai et al., 2020*; *Rogers et al., 2015*).

Our result highlights the potential importance of longer infections in the adaptation of global influenza populations, particularly where some adaptive immune response remains. A newly emergent variant under strong positive selection increases faster than linearly in frequency (*Haldane, 1924*). Given a large $N_e$, implying efficient selection, additional days of infection will have a disproportionate influence upon the potential transmission of adaptive variants. This does not imply that longer infections are the sole driving force behind global viral adaptation; selective effects affecting viral transmissibility (*Lumby et al., 2018*) would provide an alternative explanation. However, our work suggests that longer-term infections may be an important area of study in the quest to better understand global influenza virus evolution.

## Materials and methods

### Summary

In a single-locus haploid system, the expected change in a variant allele with frequency q caused by genetic drift is given by the formula (*Charlesworth, 2009*)

$$\mathrm{E}[\Delta q] = \sqrt{\frac{q(1-q)}{N_e}} \tag{1}$$

This fact has been exploited to evaluate the size of transmission bottlenecks in influenza infection, comparing statistics of genome sequence data collected before and after a transmission event (*Poon et al., 2016*; *Sobel Leonard et al., 2017b*). Such a calculation may be affected by noise in the sampling or sequencing of a population, particularly where the extent of noise outweighs the genuine change in a population (*Lumby et al., 2018*). Here we suggest that, given multiple samples from a population, an alternative approach is possible; we use this to derive a more robust estimate of $N_e$. By means of evolutionary simulations we estimate $N_e$ for cases of within-host influenza infection.

## Sequence data and bioinformatics

Sequence data describing the evolution of the infection was generated as part of a previous study (*Lumby et al., 2020*). Data, edited to remove human genome sequence data, have been deposited in the Sequence Read Archive with BioProject ID PRJNA601176. The HCV data used in validating the sequencing pipeline (see below) were previously deposited in the Sequence Read Archive with BioProject ID PRJNA380188. Processed files describing raw variant frequencies for both datasets are available, along with code used in this project, at https://github.com/cjri/FluBData (copy archived at https://github.com/elifesciences-publications/FluBData; *Illingworth, 2020a*).

Short-read data were aligned first to a broad set of influenza sequences. Sequences from this set to which the highest number of reads aligned were identified and used to carry out a second short-read alignment. The SAMFIRE software package was then used to filter the short-read data with a PHRED score cutoff of 30, to identify consensus sequences, and to calculate the number of each nucleotide found at each position in the genome. SAMFIRE is available from https://github.com/cjri/samfire (*Illingworth, 2020b*).

## Calculation of evolutionary distances

Variant frequencies at different time points during infection were used to calculate a rate of change in the population over time. We define **q**(t) as a 4 x L element vector describing the frequencies of each of the nucleotides A, C, G, and T at each locus in the viral genome at time t. We next define a distance between vectors **q**. Considering a single locus in the genome, we calculate the change in allele frequencies over time via a generalisation of the Hamming distance

$$d(q_i(t_1), q_i(t_2)) = \frac{1}{2} \sum_{a \epsilon \{A,C,G,T\}} \left| q_i^a(t_1) - q_i^a(t_2) \right| \tag{2}$$

where the term inside the sum indicates the absolute difference between the frequency of allele *a* at locus *i*. The statistic $d_i$ is equal to one in the case of a substitution, for example where only A nucleotides are observed in one sample and only G nucleotides in another. However, in contrast to the Hamming distance it further captures smaller changes in allele frequencies, lesser changes producing values between zero and one, such that a change of a variant frequency from 45% to 55% at a two-allele locus would equate to a distance of 0.1, representing half of the sum of the absolute changes in each of the two frequencies. The total distance between the two vectors may now be calculated as

$$D(\boldsymbol{q}(t_1), \boldsymbol{q}(t_2)) = \sum_i d(q_i(t_1), q_i(t_2)) \tag{3}$$

where the sum over *i* is conducted over all loci in the viral genome.

Sequence distances for non-synonymous and synonymous mutations were calculated in a similar manner, with the exception that distances were calculated over individual nucleotides rather than in a per-locus manner. We calculated

$$D^{NS}(\boldsymbol{q}(t_1), \boldsymbol{q}(t_2)) = \frac{1}{2|A_{N,i}|} \sum_{a, i \epsilon A_{N,i}} \left| q_i^a(t_1) - q_i^a(t_2) \right| \tag{4}$$

and

$$D^{S}(\boldsymbol{q}(t_1), \boldsymbol{q}(t_2)) = \frac{1}{2|A_{S,i}|} \sum_{a, i \epsilon A_{S,i}} \left| q_i^a(t_1) - q_i^a(t_2) \right| \tag{5}$$

where $A_{N,i}$ and $A_{S,i}$ are the sets of nucleotides a and positions i in the genome which respectively induce non-synonymous and synonymous changes in the consensus sequence. Synonymous and non-synonymous variants were identified with respect to influenza B protein sequences; a nucleotide substitution was defined as being non-synonymous if it induced a change in the coded protein in at least one viral protein sequence. By contrast to our primary distance measurement, values for synonymous and non-synonymous sites were calculated as mean distances per nucleotide, reflecting the differing numbers of each type of potential substitution in the viral genome.

## Estimation of effective population size

We converted our measurements of sequence distance into an estimate of $N_e$ by means of a simplified evolutionary model, assuming that all of the change in the population results from genetic drift. We first note the effect of error in measurements of the population upon our distance metric.

We suppose that at the time t, we make the observation:

$$\hat{q}(t) = \boldsymbol{q}(t) + \boldsymbol{e}(t) \tag{6}$$

where **e** is the error in measuring the population. Our definition of 'error' here is a broad one; we include both the potential for viral material in a single swab to not fully capture the entire viral diversity within the host and the potential for the sequencing pipeline to distort the composition of the material in the swab (*Illingworth et al., 2017*). In our distance calculation, we now have:

$$D(\hat{\boldsymbol{q}}(\boldsymbol{t_1}), \hat{\boldsymbol{q}}(\boldsymbol{t_2})) = \frac{1}{2} \sum_i \sum_{a \epsilon \{A,C,G,T\}} \left| (q_i^a(t_1) - q_i^a(t_2)) + \left( (e_i^a(t_1) - e_i^a(t_2)) \right) \right| \tag{7}$$

where the terms $e_i$ are locus-specific errors in the measurement of allele frequencies; we write this equation in the form:

$$D(\hat{\boldsymbol{q}}(t_1), \hat{\boldsymbol{q}}(t_2)) = D(\boldsymbol{q}(t_1), \boldsymbol{q}(t_2)) + E(\boldsymbol{q}(t_1), \boldsymbol{q}(t_2)) \tag{8}$$

where E is the deviation incurred from the true distance.

Here, given only two error-prone samples from a system, separation of the real population distance and the error term is impossible. However, given multiple samples, an approximate separation can be made. We here use linear regression to fit a model to the observed distances, fitting the model:

$$D(\hat{\boldsymbol{q}}(t_i), \hat{\boldsymbol{q}}(t_j)) \approx k|t_j - t_i| + E \tag{9}$$

for constant values k, approximating the rate of evolutionary change in the population per unit time, and E, approximating the mean amount of error in a measurement; here the term in vertical brackets is the absolute difference in time between samples i and j. This approach makes two approximations, which we believe to be either reasonable or possible to account for. Firstly, the model assumes that a linear model is appropriate to describe the change in the population over time; within our drift framework this is correct if the effective population size $N_e$ is constant, and if the distribution of allele frequencies does not change over time. In our data, the consensus population declines approximately eight-fold (*Lumby et al., 2020*), then undergoes a bottleneck due to the influence of favipiravir; we infer a representative mean value of N, selecting for clade A only samples collected before the bottleneck. Secondly, our model assumes that the deviation from truth in our distance metric does not change in a manner that is systematically associated with the time between samples. Regarding the sequencing process we believe this to be correct in so far as a consistent sequencing pipeline was used throughout. Regarding within-host population structure we note in our data a divergence over time between samples from clade A and clade B, but split these samples to obtain distinct estimates of $N_e$ for each clade. We note that large deviations from our model assumptions can be qualitatively identified by a poor fit between a simple regression model and the data.

Linear regression was performed using the Mathematica 11 software package, using the same package to calculate a 97.5% confidence interval for the calculated gradient, k.

## Wright-Fisher simulation

We next approximated the behaviour of our system using a Wright-Fisher model, re-writing the first component of *Equation 9* as

$$D(\boldsymbol{q}(t_1), \boldsymbol{q}(t_2)) \approx \Delta D(N_e, \boldsymbol{q}(t_1)) |t_2 - t_1| \tag{10}$$

Here ΔD is a stochastic function describing the change in the population, measured according to the metric D, that arises from a single generation of genetic drift in a population with effective size $N_e$ and initial allele frequencies $q(t_1)$. Regarding these allele frequencies we note that the distribution of minor allele frequencies across the genome was reasonably constant between samples for which

a good read depth was achieved (*Figure 2—figure supplement 4*; read depths for these data have previously been reported *Lumby et al., 2020*). To account for variance in these statistics we used different samples to initiate our simulations, reporting error bars across choices of $q(t_1)$.

Our Wright-Fisher model simulated the evolution of the viral population for a single generation. Rates of evolution calculated from the sequence data were rates of change per day whereas a Wright-Fisher simulation gives an estimated rate of evolution per generation. We therefore scaled the former to match the experimentally ascertained estimate of 10 hr per generation for influenza B (*Nobusawa and Sato, 2006*).

To conduct a simulation we constructed a population of N viruses. Each simulated virus had a genome comprised of eight segments, each identical in length to the corresponding segment of the influenza B virus sampled from the patient. Observations from the clinical viral population were used to specify the genetic composition of the viral population at the beginning of the simulation. A simulated population of viral genomes was established. For each viral segment, a clinical sample was chosen at random. Nucleotide frequencies at each locus in the clinical sample (modified as described below) were used to generate a multinomial sample of viruses from the simulated population, assigning alleles to viruses in the simulated population according to the random sample. This step was repeated for each locus in the segment, with no intrinsic association between alleles at different loci. The sample collected on 30th November 2017 was excluded as a starting point from this analysis due to its low read depth; all other samples had a mean read depth in excess of 2000-fold coverage.

Simulation of the population was conducted at the genome-wide level. We simulated a single generation of the evolution of our population under genetic drift, generating a random sample of N whole viral genomes from the population. Intra-segment recombination was assumed to be negligible (*Boni et al., 2008*), while reassortment between segments was neglected in line with evidence from cases of human infection (*Sobel Leonard et al., 2017a*). We collected allele frequency data from the initial and final populations, using these to calculate the distance in sequence space through which the population had evolved according to the modified Hamming distance described above.

For each population size tested, our simulation was run 400 times, using the data to produce a 97.5% confidence interval for the extent of evolutionary change at a given effective population size. For each of these 400 replicate simulations, an independent random set of samples was chosen to initiate each of the eight simulated viral segments. The extent of evolution of the real population was compared to the results from our simulated populations, giving an inference of the effective size of the viral population.

Amendments were made to the above approach.

## Accounting for false-positive variants in sequencing: Estimating a false positive rate

The evolutionary distance $\Delta D(N,q(t_1))$ calculated by our method is dependent upon the vector of allele frequencies q. Given a greater number of polymorphic alleles in a system, the evolutionary distance, calculated as the sum of allele frequency changes, will also increase. While the experimental pipeline we used has been shown to perform well in capturing within-host viral diversity (*STOP-HCV Consortium et al., 2016*), the possibility remains that sequencing could contribute additional diversity to the initial populations used in our simulation. We therefore made an estimate of the extent to which our sequencing process led to the false identification of variants. To achieve this, we used data from a previous study describing the repeat sequencing of hepatitis C virus (HCV) samples from a host (*Illingworth et al., 2017*); data in this previous study were collected using the same sequencing pipeline as that used to collect the data considered here and therefore provide a generic measure of the level of false positive variation. The data we analysed, coded as HCV01 in the original study, comprised four clinical HCV samples, each of which was split following nucleic acid extraction. Some replicate samples were processed using a DNase depletion method before all samples went through cDNA synthesis, library preparation and sequencing. DNase depletion led to samples with lower read depth; we here compared sequence data collected from the non-depleted replicates of each sample. Variant frequencies within this dataset, where variation was observed in more than one sample, are shown in *Figure 2—figure supplement 5*.

Considering the real viral sample, we note that at any given genetic locus, a minority variant either exists or does not exist according to some well-defined criterion. (For the moment the way in which variation is defined is not important; methods for defining variation, which include the use of a frequency threshold, are discussed later.) We denote the possible states of a locus as P and N, according to whether the locus is positive or negative for variation. We suppose that the probability that a random locus in the genome has a minority variant is given by $P_P$, leading to the equivalent statistic $P_N = 1 - P_P$.

Sequencing of a specific position in the genome results in the observation or non-observation of a variant. In our data we have sets of two replicate observations of each position in the genome, giving for each minority variant the possible outcomes VV, VX, XV, and XX, where V corresponds to the observation of a variant, and X corresponds to the non-observation of a variant. These observations contain errors; we denote the true positive, false positive, true negative and false negative rates of the variant identification process by $P_{V|P}$, $P_{V|N}$, $P_{X|N}$, and $P_{X|P}$ respectively. In this notation, V|P indicates the observation of a variant conditional on the variant being a true positive.

The underlying purpose of our calculation is to remove falsely detected variation from the population. We begin by assuming that the false negative rate of detecting variants is equal to zero. That is, where we do not see a variant in the sequence data, we assume that a variant is never actually present. This is a conservative step in so far as we never add unobserved variation to the population. Our assumption gives the result that the false negative rate, $P_{X|P} = 0$. In so far that a variant is never unobserved it follows that the true positive rate $P_{V|P} = 1$.

We may now construct expressions for the probabilities of observing each of the four possible outcomes. Noting that $P_{V|N} + P_{X|N} = 1$ we obtain

$$P_{VV} = P_P P_{V|P}^2 + (1 - P_P)P_{V|N}^2 = P_P + (1 - P_P)P_{V|N}^2 \tag{11}$$

$$P_{VX} = P_{XV} = P_P P_{X|P} P_{V|P} + (1 - P_P)P_{X|N}P_{V|N} = (1 - P_P)(1 - P_{V|N})P_{V|N} \tag{12}$$

$$P_{XX} = P_P P_{X|P}^2 + (1 - P_P)P_{X|N}^2 = (1 - P_P)(1 - P_{V|N})^2 \tag{13}$$

Thus the outcome probabilities may be expressed in terms of the underlying probability of a position having a variant, $P_P$, and the false positive rate $P_{V|N}$.

We next processed our sequence replicate data, considering only sites that were sequenced to a read depth of at least 2000-fold coverage. For each locus in a dataset, we calculated the observed frequency of each of the nucleotides A, C, G, and T, generating pairs which described these frequencies in each of our two replicate datasets. Removing pairs in which an allele has a frequency of more than 0.5 in either of the two datasets, we obtained a list of minority variants from each locus, generally comprising three allele frequency pairs per locus. If it is correct that two of the three minority alleles have very low frequencies, the frequencies are close to being statistically independent; the existence of a very few alleles of one minority type does not greatly affect the probability of another variant allele being observed in another read. We note that, of the more than 73 thousand sites sequenced, only 56, fewer than 0.1%, had more than one minority variant at a frequency greater than 1%. We proceeded on the assumption that each pair of minority frequencies was statistically independent of the others.

From the repeated observations of sites, we may count the number of observations of each of the four outcomes; given a total of N pairs we denote these as $N_{VV}$, $N_{VX}$, $N_{XV}$, and $N_{XX}$. Under our model of independent pairs we constructed the multinomial log likelihood of the underlying variant and false positive rates.

$$L(P_P, P_{V|N}) = \log\left[\binom{N}{N_{VV}N_{VX}N_{XV}N_{XX}}P_{VV}^{N_{VV}} P_{VX}^{N_{VX}} P_{XV}^{N_{XV}} P_{XX}^{N_{XX}}\right] \tag{14}$$

where the terms $P_{ab}$ are constructed from $P_P$ and $P_{V|N}$ according to the equations above.

Given a set of paired observations, we calculated the maximum likelihood values of $P_P$ and $P_{V|N}$. From these statistics we are able to calculate the positive predictive value of sequencing, namely the proportion of observed variants that are true positives. This is achieved by dividing the probability

that a true positive was detected (equal to the number of true positives as $P_{V|P} = 1$), by the probability that a variant was detected:

$$PPV = \frac{P_P}{P_P + (1 - P_P)P_{V|N}} \qquad (15)$$

## Frequency dependence of false-positive variant calling

Within our data, our expectation was that minority variants at higher allele frequencies would be more likely to be observed as variants in both replicate samples. We note that, where a frequency cutoff is applied to identify variants, care is required in the above protocol. For example, if a hard threshold was applied, in which variants were called at 1% frequency, a variant that was detected at frequencies of 1.01% and 0.99% would be regarded as having been observed in one case, and not observed in the other, although it likely represents a consistent observation.

In order to assess the frequency dependence of our true positive rate, we defined minimum and maximum variant frequency thresholds $q^{min}$ and $q^{max}$, and denoted the replicate observations of a minority variant frequency as $q^A$ and $q^B$ in the two samples. We further defined the frequency $q^{cut}$ according to the formula:

$$q^{cut} = \min\left\{q^{min}, \max\left\{\frac{q^{min}}{2}, 0.001\right\}\right\} \qquad (16)$$

We then defined regions of frequency space as follows:

$$VV: \begin{array}{lll} q^A \geq q^{cut}; & q^B \geq q^{cut}; & q^A + q^B \geq \frac{3q^{max}}{2}; \\ q^A max; & q^B max; & q^A + q^B < \frac{3q^{max}}{2}; \end{array}$$

$$VX: q^{min} \leq q^A max; \quad q^B cut$$

$$XV: q^A cut; \quad q^{min} \leq q^B max$$

$$XX: q^A cut; q^B cut; q^A + q^B < \frac{3q^{min}}{2} \qquad (17)$$

These inequalities are illustrated in *Figure 2—figure supplement 6*.

In the above, $q^{cut}$ functions to slightly harshen the criteria for detecting variants at low frequencies. If a variant is observed in one sample at frequency greater than $q^{min}$, then if $q^{min}$ is greater than 0.2%, the frequency in the second sample had to be at least half $q^{min}$ to be counted. If $q^{min}$ was between 0.1% and 0.2%, the frequency in the second sample had to be at least 0.1%, while if $q^{min}$ was less than 0.1%, the frequency in the second sample had to be at least $q^{min}$.

For different ranges of frequency values, $q^{min}$ and $q^{max}$, the proportion of observed variants that were true positives was calculated according to the maximum likelihood method above, using these categorisations. Results are shown in *Figure 2—figure supplement 7*. In the process of setting up the initial state of our Wright-Fisher simulation variants observed in the sequence data were considered in turn, drawing a Bernoulli random variable for each variant. Variants were included in the initial simulated population with probability equal to the proportion of observed variants that were estimated to be true positives.

## Accounting for mutation-selection balance

To account for our neglect of mutation, a frequency cutoff was applied to our simulation data. Under a pure process of genetic drift, low-frequency variants in our population are likely to die out, reaching a frequency of zero. In a real population, this would not occur, variants being sustained at low frequencies by a balance of mutation and purifying selection (*Haldane, 1937*; *Haigh, 1978*). To correct for this we post-processed the initial and final frequency values from our simulations before calculating our distance, imposing a minimum minority allele frequency of 0.1%. All changes in allele frequency below this threshold were ignored, such that, for example, if a variant changed from 0.5%

to 0%, this was processed after the fact so that the variant changed from 0.5% to 0.1%. The choice of threshold here is conservative; leading to a conservatively low estimate of $N_e$.

## Confidence intervals

Confidence intervals for the effective population size were calculated as the overlap of 97.5% confidence intervals for the evolutionary rates in the observed data, calculated from the regression for the real data, and estimated from the simulated statistics. The overlap of these values gives an approximate 95% confidence interval for $N_e$.

## Variations in methodology

A number of choices were made in our estimation of an effective population size. The effects of each of these choices were explored through further calculation and simulation. Results are shown in *Supplementary file 1*.

## Approximations in the Wright-Fisher model

In the calculation to set up an initial viral population, the assignment of minority alleles to sequences becomes slow at large population sizes. Our code simulated viral genomes; a variant allele was included into the population by choosing an appropriate proportion of genomes to which the variant was assigned. For greater computational efficiency we used a pseudo-random approach for choosing genomes. Given a population size N, we generated a set P of prime numbers that were each larger than N. Given some desired allele frequency q we wish to choose qN genomes to which to assign the variant. We therefore calculated the set of numbers:

$$a^k (mod\ p) \tag{18}$$

where p is a prime number sampled at random from the set P, and a is a randomly chosen primitive root of p. Given this choice of a and p, the values $a^k$ (where k is an integer between one and p-1) form a pseudorandom permutation of the numbers from one to p-1. We constructed a set of qN genomes by choosing genomes indexed in turn by the elements of this set, beginning from k = 1, and discarding values greater than N.

To achieve calculations for population sizes larger than $10^7$ we implemented a statistical averaging method. We generated a single population of size $10^6$, then generated 200 outcomes of a single generation of the same size, recording allele frequencies in each case. In order to simulate a value of N of size r x $10^6$ we compared the frequencies of the initial population to the mean frequencies of a random set of r outcomes. This is equivalent of simulating transmission from a population of size r x $10^6$ in which the initial population contains r copies of each of one of $10^6$ genotypes.

## Phylogenetic analysis

Consensus sequences of data were analysed using the BEAST2 software package (*Bouckaert et al., 2014*). Consensus sequences from each viral segment were concatenated then aligned using MUS-CLE (*Edgar, 2004*) before performing a phylogenetic analysis on the whole genome sequence alignment. The B/Venezuela/02/2016 sequence was used to root the alignment, the haemagglutinin segment of this virus having been identified as being very close to those from the patient. Trees were generated using the HKY substitution model (*Hasegawa et al., 1985*). A Monte Carlo process was run for 10 million iterations, generating a consensus tree with TreeAnnotator using the first 10% of trees as burn-in. Figures were made using the FigTree package (http://tree.bio.ed.ac.uk/software/figtree/).

## Haplotype reconstruction

Haplotype reconstruction was performed using multi-locus polymorphism data generated by the SAMFIRE software package (*Illingworth, 2016*). Variant loci in the genome were identified as those at which a change in the consensus nucleotide was observed between the initial and the final consensus. The short-read data were then processed, converting reads into strings of alleles observed at these loci; a single paired-end read may describe alleles at none, one, or multiple loci. Next, these strings were combined using a combinatorial algorithm to construct a list of single-segment haplotypes, sufficient to explain all of the observed data; no frequencies were inferred at this point.

Finally, a Dirichlet-multinomial model was used to infer the maximum likelihood frequencies of each haplotype given the data from each time point (*Illingworth, 2015*). Formally, we divided reads into sets, according to the loci at which they described alleles. A multi-locus variant consists of an observation of some specific alleles at the loci in question. By way of notation, we denote by $n_i^a$ the number of reads in set $i$ which describe the multi-locus variant $a$, and denote the total number of reads in the set as $N_i$. Given a set of haplotypes with frequencies given by the elements of the vector $\boldsymbol{q}$, we write as $q_i^a$ the summed frequencies of haplotypes that match each multi-locus variant $a$ in set $i$. For example, the haplotypes ATA and ATG would both match the multi-locus variant AT- describing alleles at only the first two loci. We now express a likelihood for the haplotype frequencies:

$$\mathfrak{L}(\boldsymbol{q}) = \sum_i \log \frac{\Gamma(N_i+1)}{\prod_a \Gamma(n_i^a+1)} \frac{\Gamma(\sum_a C q_i^a)}{\Gamma(\sum_a n_i^a + C q_i^a)} \prod_a \frac{\Gamma(n_i^a + C q_i^a)}{\Gamma(C q_i^a)} \tag{19}$$

Here the parameter C describes the extent of noise in the sequence data, a lower value indicating a lower confidence in the sequence data. Haplotype reconstruction was performed by finding the maximum likelihood value of the vector of haplotype frequencies $\boldsymbol{q}$. A value of C = 200 was chosen for the calculation, representing a conservative estimate given the prior performance of the sequencing pipeline used in this study (*Illingworth et al., 2017*). In contrast to previous calculations in which an evolutionary model was fitted to data (*Illingworth, 2015*), haplotype frequencies for each time point and for each viral segment were in this case inferred independently, with no underlying evolutionary model.

## Data describing influenza A/H3N2 infection

Our analysis of data describing long-term influenza A/H3N2 infection was performed on data from a previous study (*Xue et al., 2017*). As our method does not require an exceptional quality of sequencing data to calculate a rate of evolution more samples were included in our analysis than were examined in the original study. Using the codes established in the previous study, we used samples from patient W from days 0, 7, 14, 21, 28, 56, 62, 69 and 76; from patient X from days 0, 7, 14, 21, 28, 42, and 72; from patient Y from days 0, 7, 14, 21, 28, 35, 48, 56, and 70; from patient Z from days 14, 15, 20, 25, 41, 48, 55, 62, and 69. An identical procedure to that used to estimate Ne from the influenza B data was applied, calculating a rate of evolution per day from sequence data, scaling this to a rate per generation (in this case a seven hour generation time was modelled [*Nobusawa and Sato, 2006*]), and then running simulations to estimate $N_e$. We note that the estimates of false positive rate generated for the influenza B data were applied equally in this case, due to not having equivalent data to re-estimate these values. Examining the data from patient W, our distance measurements suggested potential population structure involving the samples collected on days 62 and 69; these samples were excluded from our regression analysis.

## Additional information

### Funding

| Funder | Grant reference number | Author |
| --- | --- | --- |
| Wellcome | 101239/Z/13/Z | Christopher JR Illingworth |
| Wellcome | 101239/Z/13/A | Christopher JR Illingworth |
| Wellcome | 105365/Z/14/Z | Casper K Lumby |
| Isaac Newton Trust | | Christopher JR Illingworth |
| Helsingin Yliopisto | | Christopher JR Illingworth |

The funders had no role in study design, data collection and interpretation, or the decision to submit the work for publication.

### Author contributions

Casper K Lumby, Data curation, Software, Formal analysis, Validation, Investigation, Methodology, Writing - review and editing; Lei Zhao, Formal analysis, Investigation, Methodology, Writing - review

and editing; Judith Breuer, Resources, Project administration, Writing - review and editing; Christopher JR Illingworth, Conceptualization, Resources, Data curation, Software, Formal analysis, Supervision, Funding acquisition, Validation, Investigation, Visualization, Methodology, Writing - original draft, Project administration, Writing - review and editing

**Author ORCIDs**
Casper K Lumby  http://orcid.org/0000-0001-8329-9228
Christopher JR Illingworth  https://orcid.org/0000-0002-0030-2784

**Decision letter and Author response**
Decision letter https://doi.org/10.7554/eLife.56915.sa1
Author response https://doi.org/10.7554/eLife.56915.sa2

## Additional files

### Supplementary files
• Supplementary file 1. Inferred effective population sizes for data from clade. A generated under different modelling assumptions.

• Transparent reporting form

### Data availability
All sequence data is taken from previous publications, and is available from the Sequence Read Archive. Where this is sensible, raw data underlying figures has been made available in files which accompany this document.

The following previously published datasets were used:

| Author(s) | Year | Dataset title | Dataset URL | Database and Identifier |
|---|---|---|---|---|
| Xue KS, Bloom JD | 2017 | Longitudinal deep sequencing of human influenza A (H3N2) from immunocompromised patients | https://www.ncbi.nlm.nih.gov/bioproject/PRJNA364676 | NCBI BioProject, PRJNA364676 |
| Lumby CK, Zhao L, Oporto M, Best T, Tutill H, Shah D, Veys P, Williams R, Worth A, Illingworth CRJ, Breuer J | 2020 | Favipiravir and zanamivir clear influenza B infection in an immunocompromised child | https://www.ncbi.nlm.nih.gov/bioproject/PRJNA601176 | NCBI BioProject, PRJNA601176 |

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
