## [Decision Letter]

**Acceptance summary:**

The manuscript assesses the intra-host effective population size of influenza based on longitudinal deep sequencing data from a chronic influenza B infection. Using principles modeling and statistical approaches, the authors show that the short length of a typical influenza infection is the key limiting factor upon selection at the within-host level. The topic is important, as it sheds light on the interplay between the two scales of selection within- and between-host in shaping the evolution of influenza virus.

**Decision letter after peer review:**

Thank you for submitting your article "A large effective population size for within-host influenza virus infection" for consideration by *eLife*. Your article has been reviewed by three peer reviewers, and the evaluation has been overseen by a Reviewing Editor and Aleksandra Walczak as the Senior Editor. The following individual involved in review of your submission has agreed to reveal their identity: Georgii A. Bazykin (Reviewer #2).

The reviewers have discussed the reviews with one another and the Reviewing Editor has drafted this decision to help you prepare a revised submission.

Summary:

The manuscript presents a study on within-host population genetics of influenza virus and in particular, inference of effective population size during chronic infection in immunocompromised patients. The topic is important as it explores the interplay between the two scales of selection: within-host and between-host selection that shape the evolution of influenza. Based on the analysis of sequence polymorphism, authors infer a relatively large effective population size ~10^7^ during chronic infection, in contrast to previously inferred values of ~10^2^ or less during transmission and in acute infections. All of the reviewers agree that the findings in this manuscript are interesting and a large effective population would have significant implications for efficacy of selection during within-host evolution of influenza. However, there are still some concerns regarding methodology, interpretation and presentation of the results which we would like to see addressed.

Essential revisions:

1) Comparison between chronic and acute infections:

The authors analyzed data from chronic influenza infections and concluded that the effective population size of the virus is high, including during acute infections. For instance, the authors argue that "the observed lack of within-host variation in typical cases of influenza can be explained by the short period of infection; the stochastic effects of genetic drift do not limit the impact of positive selection". It is however not evident that the authors' estimates of effective population size from chronic infections apply to acute infections given the exponential increase and decrease of viral load that dominate the course of acute infections. In fact, it's not clear that effective population size is even a very useful concept in this case.

Also, McCrone et al., 2018, and Xue and Bloom, 2020, have both shown that within-host variation in acute infections is dominated by non-synonymous mutations, and Xue and Bloom, 2020, also document stop-codon mutations within acute infections that are rarely found at appreciable frequencies in chronic infections. These observations suggest that selection is inefficient within hosts in acute infections, contrary to the authors' claims.

Moreover, McCrone et al. see radical changes in variant frequencies over the course of a few days (Figure 2E in that work) – but lineages in chronic infections (this work) persist for many months. If the authors think that N_e_ is comparable between acute and chronic infections, how do they explain the lack of diversity observed in acute infections? One way to explain this is to maintain a high N_e_ but with strong transmission bottleneck to impose stochasticity. But as point out above, "N_e_" is really not a well-defined quantity in this case. Alternatively, could the difference imply a lower census size in acute infections, and if so, is this consistent with differences in viral load? This issue is important in view of the proposed relevance of high N_e_ for long-term influenza evolution (e.g., last phrase of the Abstract and the last phrase of the Introduction).

Overall, the authors should acknowledge the differences between acute and chronic infections, and discuss their estimates in light of the previous observations. Moreover, it may also be helpful to revise the title to indicate that the manuscript focuses on chronic infections.

2) High N_e_ is inferred from small drift and a small rate of "substitutions" (which under the authors' terminology also account for minor changes in allele frequencies). In other words, the authors are inferring a large N_e_ based on the longer-term coexistence of multiple lineages within a host. Therefore, it would be important that the manuscript also discusses alternative explanations that could lead to such patterns of polymorphism. Importantly, as N_e_ in the manuscript is inferred from a Wright-Fisher (WF) model, violations in the underlying assumptions of the model can bias the results. For example, one can imagine that demographic effects like population structure could be responsible for long-term coexistence and survival of lineages, e.g., if each of the samples represents a mixture of persistent subpopulations? The authors seem to suggest this by analyzing clades A and B separately, Results and Discussion, second paragraph. Alternatively, could balancing selection in the host be responsible for maintaining this polymorphism (seems unlikely, but still a formal possibility)? A discussion and/or analysis of such alternative scenarios would be useful in assessing the robustness of the manuscript's findings.

3) Robustness of the analysis and proposed statistics:

a) It would be useful to have a clearer sense of the sensitivity of N_e_ to the cutoffs used. While a lot of care has gone into the choice, some diagrams showing the sensitivity of N_e_ to cutoff choice would better demonstrate the degree to which it is a function of low frequency variants in a straightforward way.

b) To estimate how N_e_ affects changes in allele frequencies, the authors simulate a single generation of Wright-Fisher evolution using initial allele frequencies from a randomly selected sample from the infection. As the equation in the subsection “Summary” indicates, populations with high-frequency alleles will experience larger changes in allele frequency at a given effective population size, so the initial distribution of allele frequencies from this randomly chosen sample can have a major effect on the expected change in allele frequencies. The authors show in Figure 2—figure supplement 1 that mutations can reach frequencies of 20-30% in neuraminidase, and in the influenza A patients analyzed in Figure 2—figure supplement 3, many mutations reach these and even higher frequencies, particularly at later points in the infection. The authors should run their Wright-Fisher simulations with different initial allele frequencies to evaluate how this choice of allele frequencies may affect estimates of effective population size.

c) The authors design statistic D to assess their estimation of N_e_. This statistic is a sum of changes in variant frequencies across sites (subsection “Calculation of evolutionary rates”), which is then compared between data and Wright-Fisher simulations for different N_e_ values. The authors seem to suggest that D should be more robust to noise (subsection “Summary”), without providing any evidence. In particular, the authors should clearly state how the assumptions they made about recombination structure in WF simulation could impact the statistics D and the interpretation of the inferred N_e_. From the manuscript it is not clear whether WF simulations are done at the site-wise, segment-wise, or genome-wise level, which would impact the correlation between changes in variant frequencies. For example, simulations done with high (free) recombination would expect a lower variance D compared to the case with strong linkage (data), for the same N_e_. These points should be better clarified.

4) In Figure 1A, it is clear (and the authors also mention) that the patient's viral load drops to undetectable levels for over a month of the infection, and viral load also varies substantially while the patient is continually infected. Effective population size and census population size are not always directly related, but the authors should discuss how changing population sizes affect their estimate of effective population size and whether a single effective population size is adequate to represent the infection.

5) The authors calculate sequence distance between every pair of sequenced timepoints to reduce the influence of noise from sequencing error, but as a result, the points in Figure 2A are non-independent and may contribute to a tighter confidence interval around the evolutionary rate than is realistic. In particular, changes in variant frequencies that take place during the middle of the infection will be overcounted in these pairs and will disproportionately influence the overall estimate of evolutionary rates. When the authors estimate the evolutionary distance between consecutive timepoints and divide by the number of days between them, how well does the estimate correspond to the estimates in Figure 2? What is the variance in these estimates?

6) The regression performed in Figure 2A, C, and analogous figures may be especially influenced by the few points at the right end of the distribution, which represent evolutionary distances between points spaced further apart in time. How robust is the estimate of evolutionary rate to removal of these points, or by calculation of evolutionary rate as suggested in comment 4?

7) The authors chose to infer effective population size using variants and haplotypes on the neuraminidase and hemagglutinin segments. This is an odd choice since these regions tend to experience the strongest selection, which can strongly influence the estimates of effective population size. Selection can act on linked haplotypes across the genome in some cases, but have the authors tested to see if these results hold for other gene segments as well?

8) Why are the effective population size estimates for the clade B samples calculated separately from the clade A samples? It's not evident from the SAMFIRE inference of haplotypes that clades A and B constitute separate subpopulations; it seems that they could be distinct genotypes in a well-mixed population as well, as might result from a coinfection.

9) The authors assume the generation time of 10 hours per generation for influenza B. However, if generations are longer in immunocompromised individuals, the analysis would lead to an overestimation of N_e_. Given that the main result in this manuscript is that N_e_ is high, this possibility should at least be discussed.

---

## [Author Response]

Essential revisions:1) Comparison between chronic and acute infections:The authors analyzed data from chronic influenza infections and concluded that the effective population size of the virus is high, including during acute infections. For instance, the authors argue that "the observed lack of within-host variation in typical cases of influenza can be explained by the short period of infection; the stochastic effects of genetic drift do not limit the impact of positive selection". It is however not evident that the authors' estimates of effective population size from chronic infections apply to acute infections given the exponential increase and decrease of viral load that dominate the course of acute infections. In fact, it's not clear that effective population size is even a very useful concept in this case.Also, McCrone et al., 2018, and Xue and Bloom, 2020, have both shown that within-host variation in acute infections is dominated by non-synonymous mutations, and Xue and Bloom, 2020, also document stop-codon mutations within acute infections that are rarely found at appreciable frequencies in chronic infections. These observations suggest that selection is inefficient within hosts in acute infections, contrary to the authors' claims.Moreover, McCrone et al. see radical changes in variant frequencies over the course of a few days (Figure 2E in that work) – but lineages in chronic infections (this work) persist for many months. If the authors think that N_e_ is comparable between acute and chronic infections, how do they explain the lack of diversity observed in acute infections? One way to explain this is to maintain a high N_e_ but with strong transmission bottleneck to impose stochasticity. But as point out above, "N_e_" is really not a well-defined quantity in this case. Alternatively, could the difference imply a lower census size in acute infections, and if so, is this consistent with differences in viral load? This issue is important in view of the proposed relevance of high N_e_ for long-term influenza evolution (e.g., last phrase of the Abstract and the last phrase of the Introduction).Overall, the authors should acknowledge the differences between acute and chronic infections, and discuss their estimates in light of the previous observations. Moreover, it may also be helpful to revise the title to indicate that the manuscript focuses on chronic infections.

We acknowledge that it is important to relate our result, derived from an unusual case of infection, to more regular cases of influenza in humans. We first note that what is meant in our case by the effective population size is that statistic as it relates to an established influenza infection; our data do not describe the initial founding and growth of the viral infection.

Our primary point of reference to regular influenza infection comes via measurements of CT score relating to viral infection. Our reference on this suggests about 10 fewer units of CT, or close to a 1000-fold numerical drop in census population size, in non-hospitalised, as opposed to hospitalised childhood cases. We now include the very rough calculation that this would suggest an N_e_ of around 10^4^ for such cases, cautioning that this is for an established infection, after the initial period of expansion from the transmission bottleneck.

We believe our consideration of an established population to match that of other studies of data from within-host influenza infection; in order for data to be collected from such infections, the viral population must be of some minimal consensus size. Noting that the threshold frequency at which the effect of selection outweighs that of drift is 1/N_e_s, we believe that within the window for which data can be collected, selection of 1% or greater per generation will dominate drift at an allele frequency of 1% or more. In this sense genetic drift does not limit positive selection.

On previous findings we do not completely recognise the statement that within-host variation in acute infections is dominated by non-synonymous mutations. If a simple count of variants is made, the majority will likely be non-synonymous, however this reflects a fact that the large majority of possible variants are non-synonymous for at least one viral protein. McCrone et al. state that their data, ‘suggest significant purifying selection within hosts’ while Xue and Bloom state that ‘synonymous mutations accumulate about twice as quickly as nonsynonymous mutations within hosts’. Synonymous mutations are relatively more common than nonsynonymous mutations at low frequencies, consistent with purifying selection.

We are not fully convinced that stop mutations are lethal in the traditional sense due to the nature of the influenza virus; the genome encapsulated within a virus (i.e. encoded in the RNA within a set of viral proteins) is not necessarily the same genome that was translated to produce the proteins. As such the observation of stop mutations at very low frequencies is not entirely inconsistent with efficient purifying selection. We are not aware of a great deal of work looking at stop mutations in chronic influenza infection, however the presence of purifying selection would again explain such a lack.

While we greatly admire the work of McCrone et al., we are not convinced that the within-host changes in allele frequency that they observe are caused purely by genetic drift. Selection, population structure, and rare sequencing error could all contribute to the changes observed, and the data described in that paper, with two samples collected from each individual, do not allow for discrimination between drift and these other factors. In our case, where we have multiple samples from a host, we observe both large differences between individual samples (in common with McCrone) but also an underlying pattern that suggests a large within-host population size. Previous data describing within-host evolution does not contradict our result.

We have revised the title to, ‘A large effective population size for established influenza infection’. This recognises that our inference neglects the initial phase of viral growth, which may arise from a single particle. While we cannot with our method directly evaluate N_e_ for acute infections, we believe that an argument based on CT values carries some weight when applied to these cases.

2) High N_e_ is inferred from small drift and a small rate of "substitutions" (which under the authors' terminology also account for minor changes in allele frequencies). In other words, the authors are inferring a large N_e_ based on the longer-term coexistence of multiple lineages within a host. Therefore, it would be important that the manuscript also discusses alternative explanations that could lead to such patterns of polymorphism. Importantly, as N_e_ in the manuscript is inferred from a Wright-Fisher (WF) model, violations in the underlying assumptions of the model can bias the results. For example, one can imagine that demographic effects like population structure could be responsible for long-term coexistence and survival of lineages, e.g., if each of the samples represents a mixture of persistent subpopulations? The authors seem to suggest this by analyzing clades A and B separately, Results and Discussion, second paragraph. Alternatively, could balancing selection in the host be responsible for maintaining this polymorphism (seems unlikely, but still a formal possibility)? A discussion and/or analysis of such alternative scenarios would be useful in assessing the robustness of the manuscript's findings.

For additional clarity we note that our rate is equivalent to a number of substitutions per day.

Rather than the coexistence of multiple lineages we infer a rate of N_e_ based upon the rate of change within (primarily clade A) of the viral population. Multiple lineages are not required in the sense that there could be a fully well-mixed population and we could still infer N_e_ using our method. Explicitly, we derive a rate of change in the viral population, measured across multiple samples, and identify a Wright-Fisher population which under genetic drift matches this rate of change.

We have added a few words to clarify the cladal structure of the population. We believe that the infection is founded by a single viral population (as opposed to co-infection) and that subsequently there is a branching event, so that clades A and B become spatially separated in the host and evolve independently of one another. Our guess is that the less-frequently observed clade B includes a smaller number of viruses, and so evolves faster under genetic drift.

We note two possible deviations from our Wright-Fisher model. Firstly, population structure going beyond the simple cladal structure we observe would lead to a reduction in the value of N_e_; we cite Whitlock and Barton on this point. Such population structure would alter the value we derive i.e. it will decrease N_e_ relative to a well-mixed population, leading to an increase in the rate of change of the population that our model will detect. Regarding population structure, we note that a non-well-mixed population could lead to non-representative sampling of the population and thereby increased distances between individual samples; this effect is included in our ‘error’ terminology in the method.

Secondly, we note the potential for selection to shape the population, noting the emergence of zanamivir resistance. Such selection would not be accounted for in our model i.e. to the extent that it is present it would increase the rate of change of the population that we will detect, but will attribute to a lower N_e_. In this sense the presence of positive selection would lead us to underestimate N_e_. Purifying selection is difficult to model; within the Wright-Fisher framework all selection is identical in leading to changes in allele frequencies with time. This has the consequence that as N_e_ becomes high the change in the population does not tend to zero. We note that there will be effects other than genetic drift affecting the population, and stick to our definition that our effective population size is the size at which an idealised population evolving under drift matches the behaviour of our data.

3) Robustness of the analysis and proposed statistics:a) It would be useful to have a clearer sense of the sensitivity of N_e_ to the cutoffs used. While a lot of care has gone into the choice, some diagrams showing the sensitivity of N_e_ to cutoff choice would better demonstrate the degree to which it is a function of low frequency variants in a straightforward way.

We have made two significant cutoffs in our method. The first is to remove what we believe to be false positive variant calls in the sequence data, while the second is to impose a hard cut of 0.1% allele frequency when making our calculation of distance. To evaluate these we have rerun our calculations in a way that removes each of these in turn; we find that the resulting change in N_e_ is not greatly changed by either of these. We have added Supplementary file 1 which contains inferences for calculations run with parameters other than the default parameters.

b) To estimate how N_e_ affects changes in allele frequencies, the authors simulate a single generation of Wright-Fisher evolution using initial allele frequencies from a randomly selected sample from the infection. As the equation in the subsection “Summary” indicates, populations with high-frequency alleles will experience larger changes in allele frequency at a given effective population size, so the initial distribution of allele frequencies from this randomly chosen sample can have a major effect on the expected change in allele frequencies. The authors show in Figure 2—figure supplement 1 that mutations can reach frequencies of 20-30% in neuraminidase, and in the influenza A patients analyzed in Figure 2—figure supplement 3, many mutations reach these and even higher frequencies, particularly at later points in the infection. The authors should run their Wright-Fisher simulations with different initial allele frequencies to evaluate how this choice of allele frequencies may affect estimates of effective population size.

The reviewers are correct that the allele frequencies used to initiate the Wright-Fisher model may affect the inferred effective population size. When we calculated replicate simulated populations we accounted for this; in each of the replicate simulations a random sample from the population was chosen to provide the allele frequencies for each segment of the simulated population. The uncertainty bars in our calculations therefore incorporate the uncertainty intrinsic to the initial choice of allele frequencies.

c) The authors design statistic D to assess their estimation of N_e_. This statistic is a sum of changes in variant frequencies across sites (subsection “Calculation of evolutionary rates”), which is then compared between data and Wright-Fisher simulations for different N_e_ values. The authors seem to suggest that D should be more robust to noise (subsection “Summary”), without providing any evidence. In particular, the authors should clearly state how the assumptions they made about recombination structure in WF simulation could impact the statistics D and the interpretation of the inferred N_e_. From the manuscript it is not clear whether WF simulations are done at the site-wise, segment-wise, or genome-wise level, which would impact the correlation between changes in variant frequencies. For example, simulations done with high (free) recombination would expect a lower variance D compared to the case with strong linkage (data), for the same N_e_. These points should be better clarified.

We have now incorporated further explanation into the Materials and methods, describing in a more formal manner how our statistic works and why it is more robust than a simple distance metric based upon pairs of samples from a population. We have clarified that our WF simulations were done at the genome-wise level. Based on prior evidence from human infection, simulations assumed an absence of intra-segment recombination or of reassortment between segments; this is now more clearly stated.

4) In Figure 1A, it is clear (and the authors also mention) that the patient's viral load drops to undetectable levels for over a month of the infection, and viral load also varies substantially while the patient is continually infected. Effective population size and census population size are not always directly related, but the authors should discuss how changing population sizes affect their estimate of effective population size and whether a single effective population size is adequate to represent the infection.

The calculation we make in Clade A was performed over the samples in this clade used up to the point at which favipiravir was first used; this is shown by the green box in Figure 1A. Our belief is that CT score is somewhat noisy, sometimes providing a better measurement of the amount of viral material on a swab than of the consensus population size. Previous modelling of these data suggests a smooth, roughly 8-fold decline in viral load during this period (Lumby et al., 2020). We believe that the pre-favipiravir set of samples is the most appropriate one from which to derive a headline figure for effective population size, the subsequent clinical intervention being an unusual event.

We have added a note that our estimate for clade B spanned the interval in time with the bottleneck; this may be a reason for its lower value. We also note that our estimate is of a mean effective population size.

5) The authors calculate sequence distance between every pair of sequenced timepoints to reduce the influence of noise from sequencing error, but as a result, the points in Figure 2A are non-independent and may contribute to a tighter confidence interval around the evolutionary rate than is realistic. In particular, changes in variant frequencies that take place during the middle of the infection will be overcounted in these pairs and will disproportionately influence the overall estimate of evolutionary rates. When the authors estimate the evolutionary distance between consecutive timepoints and divide by the number of days between them, how well does the estimate correspond to the estimates in Figure 2? What is the variance in these estimates?

We have provided further explanation of our method. Our basic rationale is that individual samples from the population are considerably affected by error, such that the error in each sample is larger than the true evolutionary distances undergone by the population. The raw distances between samples in consecutive timepoints are shown in the left-most points in Figure 2A; the mean of these values is 42.6 (standard deviation 8.5), with essentially no correlation between these values and the number of days which separate the samples (pvalue 0.97 from a correlation test performed in Mathematica 11). If we persist with this calculation we obtain a mean change per day in the population of just over 9 nucleotides per day, greater than the total inferred change of close to 8 nucleotides for clade A across five months of evolution. We believe that noise in the individual samples greatly outweighs the genuine signal of evolutionary change in the population in such a way that the simple comparison of pairwise samples does not produce an accurate result.

Assuming a constant underlying effective population size (or failing that, calculating some kind of mean), our regression allows us to infer a rate of evolution even in the presence of considerable noise. We acknowledge that changes in the population during the middle of infection are over-represented but do not have a solution to this; the use of multiple samples is intrinsic to our approach.

6) The regression performed in Figure 2A, C, and analogous figures may be especially influenced by the few points at the right end of the distribution, which represent evolutionary distances between points spaced further apart in time. How robust is the estimate of evolutionary rate to removal of these points, or by calculation of evolutionary rate as suggested in comment 4?

We have checked the slope of the regression by removing points from either the beginning and the end of the infection. In Figure 2A, removing between zero and four time points from each end of the data in any combination gives 25 regression coefficients; all of these fall within the 97.5% confidence interval that we report for our original calculation. [Note : Removing a time point removes all distances associated with that point, removing multiple points from the figure]. In Figure 2C there are data from only four time points; here removing either the first or the last leads to regression coefficients within the original confidence interval. We believe that the use of consecutive samples to assess effective population size gives a highly misleading result due to noise and confounding factors in the data greatly exaggerating the real rate of change of the population. We therefore omit this result from the main text, though we note that performing this calculation for clade A gives an effective population size of approximately 800.

7) The authors chose to infer effective population size using variants and haplotypes on the neuraminidase and hemagglutinin segments. This is an odd choice since these regions tend to experience the strongest selection, which can strongly influence the estimates of effective population size. Selection can act on linked haplotypes across the genome in some cases, but have the authors tested to see if these results hold for other gene segments as well?

This is a misunderstanding of our approach. We illustrate the cladal structure of the population using haplotype reconstructions calculated using sequence data for neuraminidase and haemagglutinin. These segments were chosen as they had slightly higher levels of genetic diversity, giving the clearest illustrations of a pattern that was visible across all of the viral segments. However, the calculation of effective population size was calculated genome-wide, using data from all viral segments. We have amended the text to greater highlight the illustrative nature of the haplotype reconstructions we present.

8) Why are the effective population size estimates for the clade B samples calculated separately from the clade A samples? It's not evident from the SAMFIRE inference of haplotypes that clades A and B constitute separate subpopulations; it seems that they could be distinct genotypes in a well-mixed population as well, as might result from a coinfection.

We believe that it is unlikely that these clades arise from a well-mixed population. The samples we have are deep-sequenced, generally to in excess of 2000x coverage. However, considerable differences are observed between these samples; in a well-mixed population, the samples might exhibit a pattern of evolution, but evolution would follow a continuous pattern of change rather than identifiably (in the haplotype reconstruction) being from one subpopulation or another. Our ‘clade B’ samples describe the 18^th^, 40^th^, and 41^st^ samples from the host. The 18^th^ sample is evolutionarily intermediate between everything in ‘clade A’ and the final two samples. This leads us to the belief that the two clades begin as a single transmitted population (i.e. not from a co-infection), that clades A and B are very largely spatially separate, and that clade B evolves away from clade A over time, potentially as a result of genetic drift. We have added further detail to the text describing the observed relationship between samples.

9) The authors assume the generation time of 10 hours per generation for influenza B. However, if generations are longer in immunocompromised individuals, the analysis would lead to an overestimation of N_e_. Given that the main result in this manuscript is that N_e_ is high, this possibility should at least be discussed.

We are not aware of a biological reason why the generation time for influenza would be different for immunocompromised individuals, but acknowledge that this parameter might contain some uncertainty. We have explored the effect of changes in the generation time in Supplementary file 1.